# Effect of Perioperative Blood Transfusions and Infectious Complications on Inflammatory Activation and Long-Term Survival Following Gastric Cancer Resection

**DOI:** 10.3390/cancers15010144

**Published:** 2022-12-26

**Authors:** Noelia Puértolas, Javier Osorio, Carlos Jericó, Coro Miranda, Maite Santamaría, Eva Artigau, Gonzalo Galofré, Elisenda Garsot, Alexis Luna, Aurora Aldeano, Carles Olona, Joan Molinas, Laura Pulido, Marta Gimeno, Manuel Pera

**Affiliations:** 1Service of Surgery, Hospital Universitari Mútua Terrassa, 08221 Terrassa, Spain; 2Department of Surgery, Hospital Universitari de Bellvitge, L’Hospitalet del Llobregat, 08037 Barcelona, Spain; 3Service of Internal Medicine, Hospital de Sant Joan Despí Moisès Broggi, 08970 Sant Joan Despí, Spain; 4Service of Surgery, Hospital Universitario de Navarra, 31008 Pamplona, Spain; 5Service of Surgery, Hospital Universitari Arnau de Vilanova, 25198 Lleida, Spain; 6Service of Surgery, Hospital Universitari Josep Trueta, 17007 Girona, Spain; 7Service of Surgery, Hospital de Sant Joan Despí Moisès Broggi, 08970 Sant Joan Despí, Spain; 8Service of Surgery, Hospital Universitari Germans Trias i Pujol, Universitat Autònoma de Barcelona, 08916 Badalona, Spain; 9Service of Surgery, Consorci Corporació Sanitària Parc Taulí de Sabadell, 08208 Sabadell, Spain; 10Service of Surgery, Hospital General de Granollers, 08402 Granollers, Spain; 11Service of Surgery, Hospital Universitari de Tarragona, Joan XXIII, 43005 Tarragona, Spain; 12Service of Surgery, Hospital Universitari de Vic, 08500 Vic, Spain; 13Service of Surgery, Hospital de Mataró, Consorci Sanitari del Maresme, 08304 Mataró, Spain; 14Section of Gastrointestinal Surgery, Hospital Universitario del Mar, Universitat Autònoma de Barcelona, Hospital del Mar Medical Research Institute (IMIM), 08003 Barcelona, Spain

**Keywords:** perioperative transfusion, infectious complications, inflammatory markers, gastric cancer, gastrectomy, patient blood management

## Abstract

**Simple Summary:**

Both postoperative complications and perioperative blood transfusions have been separately related to worse prognosis after gastrectomy for patients with gastric cancer. This multicenter cohort study aims to evaluate their synergic effect on inflammatory activation and prognosis. Patients were classified into four groups based on their perioperative course: one, no blood transfusion and no infectious complication; two, blood transfusion; three, infectious complication; four, both transfusion and infectious complication. The analysis shows that perioperative blood transfusion and infectious complications have a synergic effect creating a pro-inflammatory activation that favors tumor recurrence. These findings reinforce the need of promoting restrictive policies of transfusion for patients undergoing gastrectomies implementing patients blood management programs.

**Abstract:**

Background: The aim of this study was to evaluate the impact of perioperative blood transfusion and infectious complications on postoperative changes of inflammatory markers, as well as on disease-free survival (DFS) in patients undergoing curative gastric cancer resection. Methods: Multicenter cohort study in all patients undergoing gastric cancer resection with curative intent. Patients were classified into four groups based on their perioperative course: one, no blood transfusion and no infectious complication; two, blood transfusion; three, infectious complication; four, both transfusion and infectious complication. Neutrophil-to-lymphocyte ratio (NLR) was determined at diagnosis, immediately before surgery, and 10 days after surgery. A multivariate Cox regression model was used to analyze the relationship of perioperative group and dynamic changes of NLR with disease-free survival. Results: 282 patients were included, 181 in group one, 23 in group two, 55 in group three, and 23 in group four. Postoperative NLR changes showed progressive increase in the four groups. Univariate analysis showed that NLR change > 2.6 had a significant association with DFS (HR 1.55; 95% CI 1.06–2.26; *p* = 0.025), which was maintained in multivariate analysis (HR 1.67; 95% CI 1.14–2.46; *p* = 0.009). Perioperative classification was an independent predictor of DFS, with a progressive difference from group one: group two, HR 0.80 (95% CI: 0.40–1.61; *p* = 0.540); group three, HR 1.42 (95% CI: 0.88–2.30; *p* = 0.148), group four, HR 2.85 (95% CI: 1.64–4.95; *p* = 0.046). Conclusions: Combination of perioperative blood transfusion and infectious complications following gastric cancer surgery was related to greater NLR increase and poorer DFS. These findings suggest that perioperative blood transfusion and infectious complications may have a synergic effect creating a pro-inflammatory activation that favors tumor recurrence.

## 1. Introduction

Gastric cancer (GC) is one of the most common malignancies worldwide and the fourth leading cause of cancer-related death [1]. Surgery remains the main treatment for GC. Oncologic gastrectomies are complex procedures associated with high morbidity rates, ranging from 20 to 45% in Western countries [2]. Moreover, patients undergoing GC resection are at high risk of receiving blood transfusion in the perioperative period due to a high prevalence of anemia [3]. Perioperative blood transfusion leads to a higher incidence of postoperative complications, especially infectious [4]. Both postoperative complications and perioperative blood transfusions have been separately related to worse prognosis in patients with GC, probably due to postoperative cellular immune suppression and inflammatory activation [5,6,7,8]. However, there is a lack of information available to understand if the effect of transfusions and infections on inflammatory activation and tumor recurrence is independent and additive, or mutually related and synergic.

In recent years, there has been increasing concern regarding the association between the systemic inflammatory response markers and survival in patients with various types of cancer [9,10,11,12,13]. Alterations in neutrophil-to-lymphocyte ratio (NLR) and platelet-to-lymphocyte ratio have been related to higher tumor recurrence and worse long-term survival [14,15,16,17]. These inflammatory markers have mostly been determined before surgery, but some recent studies have shown that postoperative values permit a better prognostic prediction [18,19,20]. However, this pro-inflammatory activation following GC surgery has not been evaluated in relation to two potential causes: postoperative infectious complications and perioperative blood transfusions.

Patient blood management (PBM) programs include different evidence-based interventions addressed to maintain patients’ own blood mass and avoid unnecessary transfusions [21,22]. The Spanish EURECCA Esophagogastric Cancer Group recently evaluated the clinical impact of the implementation of a PBM program in gastric cancer surgery, proving a reduction in transfusion rate, infectious complications, and postoperative 90-day mortality [23]. The current prospective study aimed to investigate the dynamic changes of inflammatory markers (NLR) following curative GC surgery, their association with perioperative blood transfusion and postoperative infectious complications, and their impact on disease-free survival.

## 2. Materials and Methods

### 2.1. Study Design and Participants

A prospective multicenter study was conducted in consecutive patients undergoing elective GC resection with curative intent in 12 hospitals of the Spanish EURECCA Esophagogastric Cancer Group between January 2017 and December 2018 (ClinicalTrials.gov NCT04286984). The study was approved by the ethics committee of the Institutional Review Board of University Hospital Mutua Terrassa, and written informed consent was obtained from all patients. The study was carried out according to the guidelines of the Declaration of Helsinki. The paper has been reported in line with the STROCSS criteria [24].

Patients with non-epithelial tumors, distant metastases, evidence at diagnosis of pre-existing infections or inflammatory conditions (such as vasculitis or rheumatoid arthritis), or who were not willing to sign the informed consent, were excluded.

### 2.2. Data Collection

Clinicopathological data and follow-up status for all patients were collected from a maintained database, which was common to all institutions. Ninety variables with detailed definitions were continuously collected from each patient by the reference surgeon at each institution. Validation of data registration (period 2014–2018) in the EURECCA dataset was recently performed, revealing 97% completeness and 95% accuracy rates [25].

For each patient, recorded data included: demographics (age, sex); category of the American Society of Anesthesiologists (ASA) physical status classification system; body mass index (BMI); Eastern Cooperative Oncology Group (ECOG) performance status [26]; percentage of unintended weight loss 6 months before surgery; Charlson comorbidity index score [27], (categorized as 0–2, and ≥3); tumor location; pTNM stage (8th edition, UICC) [28]; neoadjuvant treatment; type of gastrectomy (distal subtotal or total); extension of lymphadenectomy according to the Japanese Gastric Cancer Association Classification [29]; surgical approach (open or minimally invasive); radicality, and associated multivisceral resection.

Blood samples were collected at diagnosis, immediately before and 10 days after surgery, or at the time of hospital discharge if it occurred earlier. Perioperative blood transfusion, postoperative complications, neutrophil and lymphocyte count, 30-day hospital readmission, tumor recurrence, and 90-day mortality were recorded.

### 2.3. Outcomes and Definitions

The primary endpoint of the study was to evaluate the effect of infectious complications and perioperative transfusion on disease-free survival (DFS). The secondary outcome was to evaluate their association with perioperative changes in NLR, establishing a cut-off for the change that could show an impact on DFS.

Postoperative infectious complications occurring within the first 30 days after surgery were defined according to the Gastrectomy Complications Consensus Group (GCCG) [30] and graded with the Clavien-Dindo classification [31]. Intra-abdominal infections included anastomotic leakage (defined as full thickness gastrointestinal defect involving the anastomosis), duodenal stump fistula (full thickness duodenal defect or abscess close duodenal stump), pancreatic fistula (drain output of any measurable volume of fluid with an amylase level > 3 times the upper limit of institutional normal serum amylase activity, associated with relevant symptomatology), and intra-abdominal abscess (other postoperative abnormal fluid from drainage and/or abdominal collections without gastrointestinal leaks preventing drainage removal and/or requiring treatment). Other infections were either gastrointestinal, respiratory, urinary, or other, with both symptoms and germ isolation.

Perioperative blood transfusion was defined as transfusion of allogenic red blood cells from 30 days before surgery until hospital discharge after surgery. 

In order to study both the separated and associated effects of transfusions and infectious complications on inflammatory activation and long-term outcomes, four groups of patients were defined according to their perioperative course: group one: no perioperative blood transfusion and no infectious complication; group two: perioperative blood transfusion; group three: postoperative infectious complication; group four: both blood transfusion and infectious complication. 

Disease-free survival was defined as the time from the date of surgery to recurrence of tumor or death. The latest follow-up date was March 2021.

### 2.4. Statistical Analysis

Descriptive statistics were used to summarize population characteristics. To compare the demographic and clinical profile between the 4 groups, a Chi-square test was used for categorical variables, and an ANOVA or a Kruskall–Wallis test (according to variable distribution) to compare continuous variables. NLR changes were evaluated with the ANOVA test. The paired-sample *t* test was used to determine statistically significant changes in pre- and postoperative levels of inflammatory markers. NLR change was calculated by subtracting the preoperative NLR value from the postoperative one. To dichotomize the NLR change, the R package survMisc was used to calculate the optimal cut-off point for a continuous variable when a Cox regression model is used. The Contal–O’Quigley method was used to choose the best cut-off point. The Kaplan–Meier method was used to estimate survival probabilities of the 4 perioperative groups and the 2 groups defined by the NLR change. A multivariate analysis was used to assess the effect of variates on DFS. A Cox regression model was used to identify the potentially causal effect of group on NLR and DFS. Adjustment variables were entered into a multivariate Cox regression model according to their clinical relevance. Results were expressed as hazard ratios (HRs) with 95% confidence intervals (CIs). The SPSS software package (SPSS, Chicago, IL, USA) was used to manage data and perform statistical analysis.

## 3. Results

### 3.1. Baseline Variables 

A total of 384 patients underwent GC resection with curative intent in the 12 participating hospitals between January 2017 and December 2018. After exclusion of 26 patients with metastatic disease detected at surgery, 45 patients with definitive histology of non-epithelial tumor, and 31 not having the three blood samples, 282 patients were included for final analysis. Fifty patients (17.7%) received perioperative blood transfusion. A total of 88 postoperative infectious complications were developed in 81 (28.7%) patients. Depending on their perioperative evolution, patients were grouped as follows: group one (no transfusion nor infectious complication): *n* = 181; group two (blood transfusion): *n* = 23; group three (infectious complication), *n* = 55; group four (both transfusion and infectious complication), *n* = 23. Characteristics of these four groups are summarized in Table 1. Types of postoperative complications are detailed in Table 2. Blood transfusions were associated with anemia, and cardiac and renal pathologies; infectious complications were significantly more frequent in patients who underwent total gastrectomies. Both transfusion and infectious complications were related with nodal involvement. The rest of the basal variables did not show significant differences between the four groups of patients.

### 3.2. Dynamic Changes of Inflammatory Markers

Postoperative NLR change showed a progressive increase in the four groups, ranging from +1.5 in group one to +7.7 in group four; global difference among groups was significant (*p* < 0.001), as well as between groups one and three (*p* < 0.001) and groups one and four (*p* < 0.001) (Figure 1). 

### 3.3. Effect of Postoperative Infectious Complications and Blood Transfusion on DFS

The median follow-up was 27.2 months (range 14.6–39.6). At the end of the study, 100 (35.5%) patients had died. Among the 182 patients who survived, 19 had a GC recurrence. DFS showed a progressive difference from group one to group four, as shown in Figure 2. Kaplan–Meier analysis’ log-rank test was significant (*p* = 0.0013); pairwise comparisons showed a significant difference between group one and four (*p* = 0.0004), with the rest of the differences being non-significant. In univariate analysis, the perioperative groups, the ECOG score, percentage of weight loss, surgical radicality, and pT and pN categories were found to be associated with poor DFS (Table 3). In multivariate analysis, independent prognostic factors were the perioperative group, the ECOG score, surgical radicality, and pN.

### 3.4. Effect of Inflammatory Changes on DFS

The best cut-off value for the NLR change after surgery was 2.6. As shown in Figure 3, patients with NLR change > 2.6 had worse long-term prognosis. In univariate analysis, NLR change > 2.6 had a statistically significant association with poorer DFS (HR 1.55; 95% CI 1.06–2.26; *p* = 0.025), which was maintained in multivariate analysis (HR 1.67; 95% CI 1.14–2.46; *p* = 0.009) (Table 3).

**Table 3 cancers-15-00144-t003:** Univariate and multivariate analyses of clinicopathological variables in relation to disease-free survival.

Clinicopathological Features	Univariate Analysis HR (95% CI)	*p* Ratio	Multivariate Analysis HR (95% CI)	*p* Ratio
**Perioperative classification**, (%):				
1: Trans(−)/Inf(−)	Ref.	Ref.	Ref.	Ref.
2: Trans(+)/Inf(−)	1.36 [0.70; 2.67]	0.364	0.80 [0.40; 1.61]	0.540
3: Trans(−)/Inf(+)	1.51 [0.94; 2.42]	0.086	1.42 [0.88; 2.30]	0.148
4: Trans(+)/Inf(+)	2.85 [1.64; 4.95]	<0.001	1.77 [1.01–3.11]	0.046
**NLR difference**				
<2.6	Ref.	Ref.	Ref.	Ref.
≥2.6	1.55 [1.06; 2.26]	0.025	1.67 [1.14; 2.46]	0.009
**Weight loss**				
0–5%	Ref.	Ref. (34.1%)		
6–10%	1.66 [1.08; 2.55]	0.022		
>10%	1.99 [1.20; 3.30]	0.008 (55.6%)		
**Radicality**				
R0	Ref.	Ref. (37.6%)	Ref.	Ref.
R1–R2	2.32 [1.38; 3.89]	0.001(70.8%)	2.15 [1.26; 3.69]	0.005
**pN ^a^**				
N0	Ref.	Ref. (19.7%)	Ref.	Ref.
≥N1	3.41 [2.17; 5.36]	<0.001 (56.2%)	2.89 [1.75; 4.76]	<0.001
**pT ^a^**				
≤T2	Ref.	Ref. (51.6%)	Ref.	Ref.
>T3	0.47 [0.31; 0.70]	<0.001 (25.6%)	0.87 [0.54; 1.40]	0.561
**ECOG ^b^**				
ECOG 0	Ref.	Ref. (25.8%)	Ref.	Ref.
ECOG ≥ 1	2.49 [1.59; 3.92]	<0.001 (48.1%)	2.04 [1.29; 3.24]	0.002
**ASA**				
ASA I/II	Ref.	Ref. (39.0%)		
ASA III/IV	1.25 [0.86; 1.81]	0.242 (41.8%)		

ASA, American Society of Anesthesiologists; CI, confidence interval; HR, hazard ratio; NLR (neutrophil-to-lymphocyte ratio); ^a^ According to 8th edition of the International Union Against Cancer tumor node metastasis staging system [27]; ^b^ Performance status according to Eastern Cooperative Oncology Group (ECOG) [25]^.^

## 4. Discussion

This prospective cohort study of patients undergoing GC resection with curative intent demonstrated a synergic effect of both perioperative blood transfusion and infectious complications on long-term prognosis. Previous studies have separately studied the impact of complications and blood transfusion following GC resection on DFS, with heterogeneous results: some studies support a negative effect of anastomotic leakage, postoperative infection, or intra-abdominal infection in prognosis [32,33,34,35], while others did not find that relation [36]; moreover, the adverse prognostic effect of perioperative blood transfusion has also been confirmed in many studies [37,38,39,40], including some systematic reviews and meta-analyses [41,42,43,44], while other studies did not find this association [45,46,47]. To date, only two studies, one of patients with colorectal cancer [48], and another with locally advanced gastric cancer from Xiao et al. [49] (with no determination of inflammatory markers), have analyzed the combined effect of perioperative blood transfusion and postoperative infectious complications, suggesting their additive negative association with worse cancer specific survival. In univariate analysis of the present study, perioperative transfusions and infections could also seem to have an additive effect on DFS, as the sum of their HR is like the one deriving from their combination. However, looking at the results of multivariate analysis, it seems more likely to think of a synergetic effect of both variables, which would potentiate their individual effect on DFS. If we consider that both variables are also causally related, as perioperative blood transfusion can also favor postoperative infections, our findings support the need for promoting PBM policies in GC surgery, in order to minimize unnecessary transfusions [50].

Inflammatory ratios showed a postoperative increase that was significantly greater when perioperative blood transfusion and infectious complications occurred simultaneously. In addition, NLR postoperative change > 2.6 showed a good predictive accuracy for determining DFS. This finding could have some clinical implications, as patients with a higher postoperative NLR increase may potentially benefit from adjuvant treatment and/or a closer long-term follow-up. Various studies have confirmed the prognostic usefulness of preoperative inflammatory markers in patients with GC, with different cut-off points [15,16,51,52,53]. Those studies focused only on the initial values and not on the dynamic changes after surgery. In contrast, other authors have reported that postoperative inflammatory markers, especially NLR, behaved as better prognostic predictors [18,19,20]. A recent study retrospectively analyzed the dynamic changes of inflammatory markers in patients who underwent radical gastrectomy, showing a significantly improved prognostic accuracy by incorporating the post-12-month lymphocyte–monocyte ratio in the TNM staging system [19]. However, in all those studies, postoperative inflammatory activation was not related to its most probable causes: perioperative blood transfusion and postoperative infectious complications.

This study has some limitations, including its reduced sample size, particularly in perioperative groups two and four. Prospective studies with more patients will help determine the relationship of postoperative adverse events with inflammatory activation and prognosis with greater precision. In addition, the lack of previous consensus on the optimal cut-off value for inflammatory markers has made comparison of results difficult. Due to the growing evidence of the prognostic relevance of inflammatory markers, an effort is needed to establish their optimal categories in order to assist in clinical decision making. Additionally, heterogeneity in the chronological definition of perioperative blood transfusion in the literature adds difficulty to benchmarking.

## 5. Conclusions

In conclusions, this study shows that blood transfusion and infectious complications may have a synergic impact on short-term increase in NLR and poorer long-term outcomes in patients undergoing gastric cancer surgery. As infectious complications are favored by perioperative transfusions, these findings reinforce the need of promoting PBM policies in order to minimize pro-inflammatory activation immediately after surgery that may favor tumor recurrence.

## Figures and Tables

**Figure 1 cancers-15-00144-f001:**
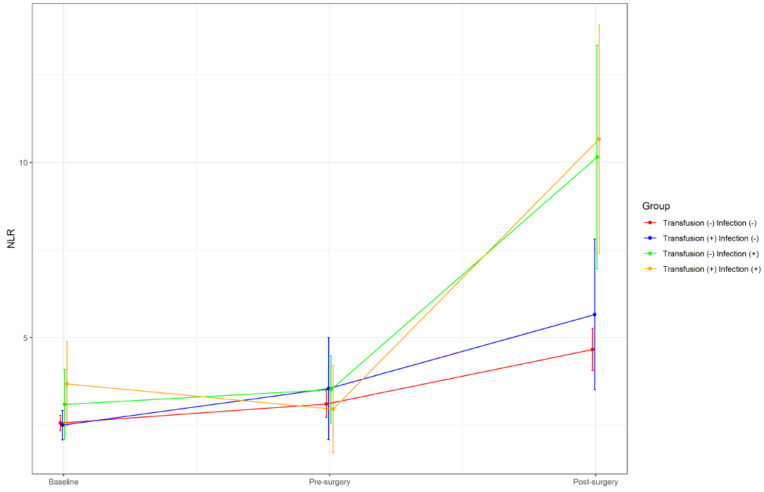
Perioperative neutrophil-to-lymphocyte ratio (NLR) changes classified in four groups depending on their perioperative course. Values are expressed as mean and standard deviation.

**Figure 2 cancers-15-00144-f002:**
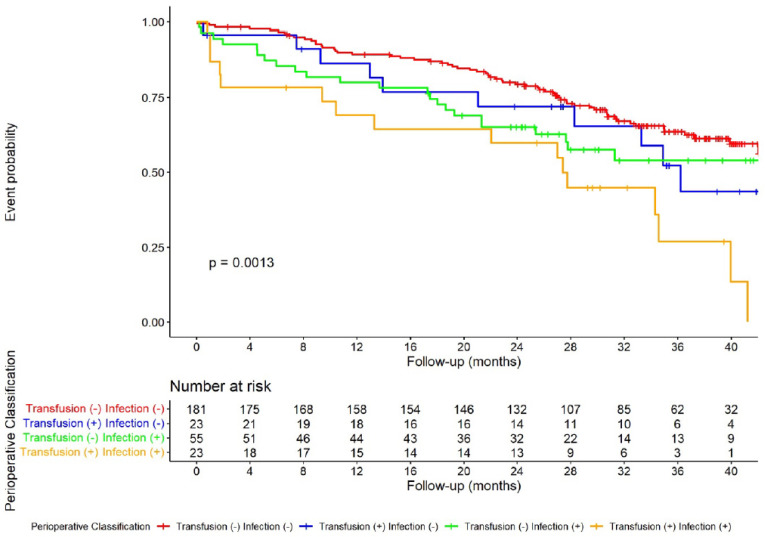
Kaplan–Meier analysis of probability of disease-free survival according to the perioperative classification.

**Figure 3 cancers-15-00144-f003:**
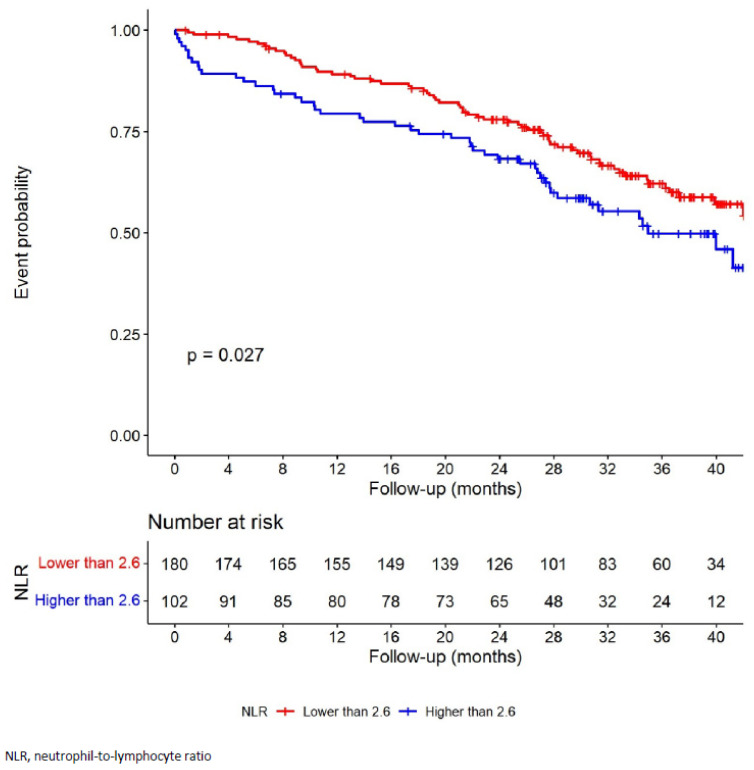
Kaplan–Meier analysis of probability of disease-free survival according to the NLR change cut-off.

**Table 1 cancers-15-00144-t001:** Demographics, comorbidities; analytical, surgical, and pathology characteristics in the study population, classified in four groups depending on their perioperative course.

Variables	Transfusion (−) Infection (−)	Transfusion (+) Infection (−)	Transfusion (−) Infection (+)	Transfusion (+) Infection (+)	*p* Values
	N = 181	N = 23	N = 55	N = 23	
Age, Mean (SD)	68.4 (11.5)	76.0(9.32)	70.8 (12.3)	74.4(8.27)	0.004
Sex, N (%):					0.059
Male	118 (65.2%)	9 (39.1%)	39 (70.9%)	15 (65.2%)	
Female	63 (34.8%)	14(60.9%)	16 (29.1%)	8 (34.8%)	
ASA score, N (%):					0.237
ASA I	7 (3.87%)	0 (0.00%)	0 (0.00%)	0 (0.00%)	
ASA II	85 (47.0%)	7 (30.4%)	28 (50.9%)	9 (39.1%)	
ASA III	85 (47.0%)	15(65.2%)	27 (49.1%)	12 (52.2%)	
ASA IV	4 (2.21%)	1 (4.35%)	0 (0.00%)	2 (8.70%)	
ECOG ^c^, N (%):					0.027
ECOG 0	70 (38.7%)	4 (17.4%)	20 (36.4%)	3 (13.0%)	
ECOG ≥ 1	111 (61.3%)	19(82.6%)	35 (63.6%)	20 (87.0%)	
Charlson score ^a^, N (%):					0.376
0–2	98 (54.1%)	10(43.5%)	25 (45.5%)	9 (39.1%)	
≥3	83 (45.9%)	13(56.5%)	30 (54.5%)	14 (60.9%)	
Ischemic heart, N (%): disease, N (%)	6 (3.31%)	3 (13.0%)	1 (1.82%)	2 (8.70%)	0.076
CHF, N (%)	5 (2.76%)	3 (13.0%)	1 (1.82%)	3 (13.0%)	0.017
DM complic., N (%)	26 (14.4%)	4 (17.4%)	13 (23.6%)	7 (30.4%)	0.138
DM no complic., N (%)	12 (6.63%)	2 (8.70%)	3 (5.45%)	2 (8.70%)	0.840
COPD, N (%)	22 (12.2%)	3 (13.0%)	12 (21.8%)	4 (17.4%)	0.298
Renal failure, N (%)	5 (2.76%)	1 (4.35%)	1 (1.82%)	4 (17.4%)	0.019
Vasc. Dis., N (%)	20 (11.0%)	5 (21.7%)	9 (16.4%)	2 (8.70%)	0.350
Weight loss, N (%):					0.174
0–5%	120 (66.3%)	17(73.9%)	36 (65.5%)	12 (52.2%)	
6–10%	41 (22.7%)	1 (4.35%)	11 (20.0%)	8 (34.8%)	
>10%	20 (11.0%)	5 (21.7%)	8 (14.5%)	3 (13.0%)	
Hb (g/dL), Mean (SD)	12.0 (2.63)	10.4(2.26)	11.8 (2.57)	10.8 (2.99)	0.012
Tumor location, N (%):					0.060
Upper third	15 (8.29%)	1 (4.35%)	10 (18.2%)	4 (17.4%)	
Middle third	67 (37.0%)	6 (26.1%)	21 (38.2%)	7 (30.4%)	
Lower third	97 (53.6%)	16 (69.6%)	23 (41.8%)	12 (52.2%)	
Entire	2 (1.10%)	0 (0.00%)	1 (1.82%)	0 (0.00%)	
Neoadjuvancy, N (%)	70 (38.7%)	4 (17.4%)	25 (45.5%)	10 (43.5%)	0.128
Gastrectomy, N (%):					0.006
Distal subtotal	117 (64.6%)	18(78.3%)	23 (41.8%)	13 (56.5%)	
Total	64 (35.4%)	5 (21.7%)	32 (58.2%)	10 (43.5%)	
Access, N (%):					0.269
Open	88 (48.6%)	14(60.9%)	23 (41.8%)	8 (34.8%)	
Laparoscopic	93 (51.4%)	9 (39.1%)	32 (58.2%)	15 (65.2%)	
pT ^b^, N (%):					0.179
≤T2	98 (54.1%)	17(73.9%)	30 (54.5%)	16 (69.6%)	
>T3	83 (45.9%)	6 (26.1%)	25 (45.5%)	7 (30.4%)	
pN ^b^, N (%):					0.008
N0	91 (50.3%)	6 (26.1%)	20 (36.4%)	5 (21.7%)	
≥N1	90 (49.7%)	17(73.9%)	35 (63.6%)	18 (78.3%)	
Node count, Mean (SD)	27.6 (15.8)	27.7(16.3)	26.5 (13.5)	27.7 (9.69)	0.972
Radicality, N (%):					0.881
R0	164 (90.6%)	21(91.3%)	52 (94.5%)	21 (91.3%)	
R1–R2	17 (9.39%)	2 (8.70%)	3 (5.45%)	2 (8.70%)	

ASA, American Society of Anesthesiologists; CHF, congestive heart failure; DM, diabetes mellitus; COPD, chronic obstructive pulmonary disease; Vasc. Dis., vascular disease; SD, standard deviation. ^a^ According to Charlson Comorbidity Index [26]; ^b^ According to 8th edition of the International Union Against Cancer tumor node metastasis staging system [27]; ^c^ Performance status according to Eastern Cooperative Oncology Group (ECOG) [25].

**Table 2 cancers-15-00144-t002:** Type of postoperative infectious complication, as defined according to the Gastrectomy Complications Consensus Group (GCCG) [29]. A total of 88 postoperative infectious complications in 81 patients.

lnfectious Complications ^a^	*n* (%) ^b^
Gastrointestinal complications	
Anastomotic leak	24 (8.3%)
Duodenal stump leak	7 (2.4%)
Pancreatic fistula	2 (0.7%)
Abdominal collection	17 (5.5%)
Clostridium infection	1 (0.3%)
Wound infection	7 (2.4%)
Pneumonia	13 (4.5%)
Catheter infection	10 (3.4%)
Urinary infection	7 (2.4%)

^a^ There were seven patients with two postoperative infectious complications; ^b^ % is given above total of patients included in the study.

## Data Availability

The data presented in this study are available on request from the corresponding author.

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
