# Peer review of "Effect of Perioperative Blood Transfusions and Infectious Complications on Inflammatory Activation and Long-Term Survival Following Gastric Cancer Resection"

_cancers, 2022, doi:10.3390/cancers15010144_

Round 1

Reviewer 1 Report (Previous Reviewer 1)

I would thank the Authors for meticulously addressing all my comments. The scientific soundness of the manuscript has substantially improved. I feel the conclusions are now adequately supported by the results.

In my opinion, the manuscript can now be accepted in its current form. 

This manuscript is a resubmission of an earlier submission. The following is a list of the peer review reports and author responses from that submission.

Round 1

Reviewer 1 Report

I would like to thank the Editor for giving me the opportunity to review this interesting manuscript.

In this prospective multicenter study, the Authors investigated the impact of perioperative blood transfusion and infectious complications on postoperative variations of inflammatory markers and DFS in cancer patients undergoing gastrectomy with curative intent. Although the study seems not to add a true novelty to the current literature on the topic, I think it may represent an interesting read, especially for the identification of a cutoff value for NLR that allows a valid DFS prediction, and the Authors should be commended for their efforts.

Please find below my comments.

1)      In my opinion, the conclusion drawn by the Authors regarding blood transfusion and infectious complications should be reconsidered. The Authors stated that these two variables are additive/synergic. Additive and synergic are not synonyms: an additive effect refers to a combination that provides the sum of the effects of the individual components; a synergistic effect occurs when the effect is greater than the sum of individual components (https://doi.org/10.3389/fphar.2017.00158).

Looking at table 3, at a first glance, the two variables seem to have an additive effect in univariate analysis (the sum of the HR is quite identical to the HR deriving from their combination). But looking at multivariate analysis the assumption of an additive effect cracks. The Authors should revise the conclusions accordingly.

Furthermore, HRs and the relative 95% CI for group 4 are exactly the same in univariate- and multivariate analysis. Is that correct? Please verify and modify in case of error.

2)      Since future perspectives could be very interesting, I would suggest the Authors expand a bit more in the discussion about the potential clinical and therapeutic implications of their findings (especially the NLR cutoff).

Minor comments

-          Abstract: I would suggest avoiding “we” while presenting in scientific papers

-          Line 140: Since independent- and paired-samples t-test was adopted to compare means, I presume the assumption of normality was not violated. If so, please, state it in the methods. If not, the choice of a parametric test to compare continuous variables among groups would not be appropriate.

-          Line 142: I would suggest better defining how the NLR cutoff was identified. SurvMisc is an R package, not a statistical method. I am an R user, so I know what the package does, but most of the readers are not familiar with this statistical environment.

-          By the end of the study 100 patients have died and 19 experienced disease recurrence. In the multivariate Cox proportional hazards model, 8 variables were included, which is appropriate for the number of events. Nevertheless, it is unclear how was the model built: for instance, if a forced entry method was used (rather than a stepwise backward selection), no mention of multicollinearity and overfitting assessment was made in the methods section. Please specify.

-          Line 162-166: the authors state “Blood transfusions were associated with anemia, cardiac and renal pathologies; infectious complications were significantly more frequent in patients who underwent total gastrectomies. Both transfusion and infectious complications were related with nodal involvement. The rest of basal variables did not show significant differences between the four groups of patients”. However, the results of these comparisons (p values) are not reported in the text, nor in the tables. Please correct.

-          Line 180: I would suggest the Authors specifying that the NLR change refers to the pre- /post-surgery variations rather than with NLR baseline values.

-          Figure 2 shows K-M curves for DFS of the 4 groups. Although the log-rank test is significant (p = 0.0013) no mention of pairwise comparisons is made in the text nor in the figure. In other words, there’s an evident difference (at least graphically) between groups 1 and 4; but what about between 2 and 3, or 3 and 4? Please specify.

-          Lines 256-258: “In conclusion, our study shows that both blood transfusion and infectious complications have an independent and additive impact on short-term increase in NLR and poorer long-term outcomes in patients undergoing gastric cancer surgery.” Actually, as a result of multivariate analysis, these variables do not have an independent effect on DFS. I guess the Authors intended that blood transfusion and infectious complications have an additive effect and their combination represents an independent predictor of DFS. Please rephrase the sentence.

-          Minor English language corrections are recommended, mainly in the abstract.

Reviewer 2 Report

In this study Puertolas et al., reported the results of a retrospective analysis regarding the effects of perioperative blood transfusion and infectious complications on postoperative inflammatory markers and DFS. The manuscript is well written and utilizes the appropriate methodology. However there are some major issues that need to be adressed by the authors:

- The authors utilized a 4-group analysis of the available sample (no transfusion no SSI, transfusion no SSI, no transfusion SSI, transfusion SSI). However, based on current literature, transfusion is a risk factor for the development of postoperative SSI. Therefore study design should be re-considered since there is an inherent relation between the analyzed subgroups.

-Consider utilizing a matching algorithm (i.e. PSM) for the compared study subgroups. This would enhance the validity of the reported results

-Consider reporting the p values of all base demographics and NLR comparisons

- Multivariate analyses showed that only the transfusion and SSI subgroup had a significant effect on DFS. However it is widely accepted that both SSI and transfusion are related to postoperative survival outcomes. Please comment on the discrepancy of these findings.
